ecology, evolution

environmental variability, source–sink system, experimental evolution, host adaptation, stepping stone

**Author for correspondence:**
Alicja Laska
e-mail: alicja.laska@amu.edu.pl

†These authors contributed equally to this study.

# A sink host allows a specialist herbivore to persist in a seasonal source

Alicja Laska[1], Sara Magalhães[2], Mariusz Lewandowski[3], Ewa Puchalska[3], Kamila Karpicka-Ignatowska[1], Anna Radwańska[1], Shawn Meagher[4], Lechosław Kuczyński[1,†] and Anna Skoracka[1,†]

[1]Population Ecology Laboratory, Institute of Environmental Biology, Faculty of Biology, Adam Mickiewicz University, Uniwersytetu Poznańskiego 6, 61-614 Poznań, Poland
[2]cE3c, Centre for Ecology, Evolution and Environmental changes, Faculdade de Ciências, Universidade de Lisboa, Campo Grande, Edifício C2, 1749-016 Lisboa, Portugal
[3]Section of Applied Entomology, Department of Plant Protection, Institute of Horticultural Sciences, Warsaw University of Life Sciences − SGGW, Nowoursynowska 159, 02-787 Warsaw, Poland
[4]Department of Biological Sciences, Western Illinois University, Macomb, IL 61455, USA

 

AL, 0000-0002-6743-6015; ML, 0000-0001-5360-0382; EP, 0000-0001-6032-3571;
KK-I, 0000-0002-1071-5742; AR, 0000-0001-5464-3659; LK, 0000-0003-3498-5445;
AS, 0000-0002-9485-532X

In seasonal environments, sinks that are more persistent than sources may serve as temporal stepping stones for specialists. However, this possibility has to our knowledge, not been demonstrated to date, as such environments are thought to select for generalists, and the role of sinks, both in the field and in the laboratory, is difficult to document. Here, we used laboratory experiments to show that herbivorous arthropods associated with seasonally absent main (source) habitats can endure on a suboptimal (sink) host for several generations, albeit with a negative growth rate. Additionally, they dispersed towards this host less often than towards the main host and accepted it less often than the main host. Finally, repeated experimental evolution attempts revealed no adaptation to the suboptimal host. Nevertheless, field observations showed that arthropods are found in suboptimal habitats when the main habitat is unavailable. Together, these results show that evolutionary rescue in the suboptimal habitat is not possible. Instead, the sink habitat functions as a temporal stepping stone, allowing for the persistence of a specialist when the source habitat is gone.

## 1. Introduction

Most organisms face temporal and spatial environmental variability. In extreme cases, environmental variation may lead to certain habitats becoming temporally entirely unavailable, for instance, owing to strong seasonality, pulsed resources or human activity [1–4]. Nevertheless, many species occur in these highly variable environments [5], which necessitates an understanding of the mechanisms that allow their persistence.

Over ecological time, species distributions in temporally variable environments are generally not restricted to habitat patches where population growth is positive. Local demographic processes coupled with dispersal often lead to source–sink dynamics, in which persistence in sinks (where the population growth rate is negative) is contingent upon immigration from sources, i.e. areas with positive growth [6–9]. Even though sinks cannot persist in isolation, they may lead to the stabilization of the overall demographic system and, therefore, guarantee the long-term persistence of populations [8,9]. For example, sinks can act as alternative habitats when sources are overcrowded [6,7,10,11] or temporally absent.

Evolution in sinks (i.e. evolutionary rescue) is possible under specific migration and local adaptation values [12,13], population sizes [14], rates of environmental

change [15] or temporal fluctuations in the availability of the sink habitat [16]. Empirical tests have revealed that adaptation to sink habitats occurs in some cases (e.g. [17,18]) but not in others (e.g. [19]). If evolutionary rescue occurs in sinks in temporally variable environments, then theory predicts the evolution of generalists, which perform reasonably well (i.e. have a positive growth rate in the long run), across a wide range of habitats [20–23]. This prediction has been confirmed in some empirical studies (e.g. [24,25]) but not in others (e.g. [26–28]). Additionally, other responses, such as adaptation to habitat switching [29] or diversification among populations [28,30,31], can evolve in temporally variable environments.

To fully understand species distribution and performance in source–sink environments, these ecological and evolutionary perspectives should be merged in a common framework [32]. Such studies, however, are rare, owing to incomplete fitness measures, the difficulty of identifying sinks in the field or limited study durations [32]. Here, we aimed to fill this gap and applied a comprehensive ecological-evolutionary approach to explain the persistence of organisms in temporally unavailable habitats. Obligatory herbivorous arthropods provide ideal study systems for this approach because most herbivorous crop pests are associated with temporally fluctuating habitats. They typically occur on hosts that are harvested at a particular time of the year and hence are temporally unavailable. Some studies have demonstrated the occurrence of spatial source–sink dynamics and their impact on herbivore populations (e.g. [33–37]). However, studies on the effect of temporal sinks on herbivorous arthropods are much scarcer (but see [38,39]).

Here, we address how a herbivorous wheat curl mite (*Aceria tosichella*, WCM hereafter) copes with its cyclically fluctuating environment, in agricultural systems (cereal fields). WCM is a crop pest of wheat, but it also occurs on other cereals and permanently available wild grasses.

Our goals were to determine whether this arthropod is truly a generalist by rigorously evaluating whether its habitats are sources or sinks, determine whether evolutionary rescue is possible on a sink host, and document seasonal patterns of host use to test for evidence of temporal source–sink population dynamics. To this aim, we first estimated population growth in as well as acceptance of and emigration towards both habitats. Second, we performed an experimental evolution study in a potential sink or a source, either with or without temporal variation between these habitats. Finally, we documented occupancy in both habitats for several months in the field. If WCM is a true generalist, we expect that (i) it will have positive population growth in both habitats for many generations, (ii) it will disperse towards good and poor habitats at a similar rate and it will readily accept both habitats, (iii) it will adapt to the sink habitat during long-term experimental evolution, and (iv) its prevalence in the field will exclusively hinge upon host availability. By combining laboratory experiments (encompassing population growth, behaviour and experimental evolution) and field surveys, we demonstrated an important role of the sink habitat in temporally varying environments.

## 2. Material and methods

### (a) The study system
The WCM has long been considered a generalist phytophagous mite found on approximately 100 species of grasses [40]. However, DNA barcoding has indicated that WCM actually represents a cryptic species complex consisting of at least 29 genetically divergent lineages that differ in their host specificity [41]. Here, we conducted experiments on the WCM MT-1 genotype (known as type 2 in Australia and North America, [42,43]), which is distinguished by its distinct mitochondrial cytochrome C oxidase subunit I (COI) sequence [44]. This genotype exhibits a very high population intrinsic growth rate on wheat (*Triticum aestivum* L.) ($R_0 = 50.5$; 95% confidence interval (CI) = 46.2–57.1), but it also develops on other cereals and wild grasses, albeit at lower rates. On smooth brome (*Bromus inermis* Leyss.), a wild grass species used in this study, intrinsic population growth is six times lower ($R_0 = 8.2$; 95% CI = 7.7–8.9) than that on wheat [45].

### (b) Experiments
#### (i) Mite stock population
For all experiments, we used WCM MT-1 individuals from a genetically diverse stock population that was established in November 2017 using WCMs collected from nine localities in Poland (1–5 populations from each locality). Field-derived (i.e. initial) populations were initiated with 1–100 individuals that had been collected from a separate wheat spike or grain (*ca* 500 total wild individuals). During the build-up of the initial populations, randomly chosen individuals were barcoded (using COI) to confirm their MT-1 genotype. Once the initial populations were established (in total 26), approximately 1000 adult females from each of the populations were combined to establish the stock population. The stock population was maintained for four weeks before individuals were used in the experiment. All populations were maintained on wheat plants under constant conditions (22–24°C, 12 L : 12 D cycle; 40% relative humidity (RH)). Plants for all populations and experiments were grown from seeds and cultivated in pots in separate rooms to avoid accidental mite infestation. Details of population creation and husbandry conditions can be found in the electronic supplementary material, appendix S1.

#### (ii) Population growth rate
To determine whether wheat and brome were source and sink environments, respectively, we assessed the WCM population growth rate on these plants. For this purpose, approximately 300 mites were transferred from the stock population to clean potted wheat or smooth brome plants (10–14 and 30 days old, respectively, corresponding to approximately the same biomass: leaves of at least *ca* 100 mm long and 5 mm width). The plants (20 per pot, 30 pots per species) with mites were kept in incubators under controlled conditions (27°C, photoperiod 16 L : 8 D cycle, 60% RH). Egg-to-egg developmental time of WCM MT-1 at 27°C is 7 days [46]; thus, we counted mites after 14, 21 and 28 days, roughly corresponding to two, three and four generations, respectively, with 10 replicates (i.e. single pots) of each plant species per time interval. Because accurately counting the number of mites required destructive sampling, we used 30 pots per host species to obtain 10 replicates per plant species and time interval.

#### (iii) Emigration
WCMs, as all herbivorous mites, disperse passively with wind, and thus the place where such dispersers land is outside their control [47–49]. Owing to such unpredictability, the decision to initiate and undertake dispersal is especially crucial. Indeed, there is ample evidence that herbivorous mites use different types of cues upon which they base their decision to undertake aerial dispersal [50,51]. Moreover, they are also ambulatory dispersers, using cues to move from or towards patches [52,53]. Here, we evaluate these two types of dispersal in WCM exposed to wheat or brome.

Emigration was measured as (i) dispersal rate via wind from wheat towards both hosts (the proportion of individuals that dispersed relative to the total number of individuals on the source plant), and (ii) acceptance rate of mites placed on both hosts (the proportion of individuals that stayed on the experimental arena relative to the total number of individuals placed on it).

Wind dispersal was measured in wind tunnels built according to Kuczyński et al. [54]. These wind tunnels were composed of (i) a wind generator producing wind at the speed of 2.5 m s$^{-1}$ (sufficient wind speed for WCM MT-1 dispersal; [55]); (ii) source plants, i.e. wheat plants infested with mites that were exposed to wind to trigger mite dispersal; and (iii) target plants, i.e. an area composed of either brome or wheat plants on which mites could settle after dispersal. Mites were exposed to a fluctuating wind regime (electronic supplementary material, appendix S2) to mimic natural conditions in which the wind blows intermittently and allow mites to receive cues (kairomones) from the plant located downwind. Single upwind source plants were transplanted from the mite stock population and contained 1000 to 3500 mites. WCM densities on source plants were similar in treatments with both target plant species and had no effect on dispersal rates (electronic supplementary material, appendix S3). Each blowing session lasted 24 h. After exposure to wind currents, the number of individuals on the source plant was counted. The number of dispersers was estimated as the difference in the number of mites present on source plants before and after each blowing session. There were 10 replicates per treatment (wheat–wheat or wheat–brome).

To measure host acceptance, 10 adult WCM females were placed on leaf fragments (5 × 5 mm) of either wheat or brome in individual wells of 6-well Plexiglas plates filled with artificial culture medium [56]. After 30 min, the number of mites that remained on the leaf fragments was counted. There were 20 replicates for each host plant species.

### (iv) Experimental evolution
WCM MT-1 has been found on brome in the field. Moreover, laboratory experiments have shown that its growth rate after two generations allows population replacement [45]. Therefore, we tested whether WCM MT-1 adapted to this plant species using an experimental evolution set-up. Mites from the stock population were allocated to three host–plant selection regimes: wheat (10 populations), brome (33 populations) and wheat–brome (alternating every three generations on each host species; 30 populations). Each replicate population was established with approximately 300 WCM MT-1 individuals transferred from the stock population to potted plants (20 plants per pot). Independent regimes were incubated separately in growing chambers (under the same conditions as the population growth rate experiment). Every three weeks (three WCM generations at 27°C), approximately 300 individuals from each population were transferred to a pot with 20 new plants according to the selection regime. Populations on wheat evolved for 39 generations (and they are still being maintained, currently reaching approximately 100 generations with no extinctions observed), whereas those on the other regimes were followed until extinction.

### (c) Seasonal pattern of host plant infestation
To characterize the temporal patterns of natural infestation on wheat and brome, we used data collected from the entire area of Poland (greater than 311 000 km$^2$) during the summer season from June to August 2012–2014. To achieve an even distribution of sampling localities, a stratified random sampling scheme was used. The area of Poland was divided into 367 30 × 30 km squares (i.e. strata). Within each stratum, a 1 × 1 km square of the agrarian landscape was randomly selected. Randomization was restricted to agrarian cover types based on the Corine Land Cover database

[57]. At the centre of each 1 × 1 km square, wheat from the cereal field and smooth brome from nearby field margins were collected. Each stratum was sampled once during the season. To collect mature cereals in the period they are available (two months), the area of Poland was divided into four parts and in each part, collections were made simultaneously by a different group of researchers. Each sample consisted of at least 10 plant shoots of a given plant species (total samples: 281). Samples were transported to the laboratory where each entire plant (leaves, leaf sheathes and seed head spikes) was examined under a stereomicroscope (for more details see [58]). The number of WCMs was recorded, after which mites were soaked in ATL tissue lysis buffer for subsequent DNA identification using COI barcodes (according to [59]).

DNA identification of the 2012–2014 surveys showed the total absence of the WCM MT-1 genotype on brome in locations where the main host (wheat) was available in the field. Other WCM genotypes specialized to brome, namely MT-9, MT-10 and MT-14 [41] were found on brome in these locations. To assess mite distribution in other periods, we used additional data over a longer period of time (2007–2014), covering the time before wheat ripening and after wheat harvesting (i.e. April to October). These data included 37 samples in total (all collected from different locations) consisting of at least five shoots. Samples were collected and examined, WCM specimens were counted, and DNA was barcoded as described above. The distribution of all 318 sampling locations is presented in the electronic supplementary material, appendix S4.

Data were not collected from November (late autumn) to March (early spring) because fields are often hidden under snow cover, and thus mites are very difficult to find.

### (d) Statistical analyses
All statistical analyses were performed in R v. 4.0, [60] using the 'mgcv' package to fit generalized additive mixed models (GAMMs) [61] and the 'glmmTMB' package to fit generalized linear models (GLMs) [62].

### (i) Population growth rate
*The per capita* population growth rate (*r*) was used as a measure of the reproductive performance of WCM on both host plant species. This was defined according to the following formula:

$$r = \ln\left(\frac{n}{n_0}\right)/t \,,$$

where $n_0$ corresponds to the number of females placed on each plant at the beginning of the experiment, and *n* corresponds to the number of mites (which are progeny of the $n_0$ females) after each tested time period, *t*, where *t* was expressed as the number of generations. If $r < 0$, the population size decreases and $r = 0$ indicates no change in population size.

To test whether the population growth rates differed between wheat and smooth brome and changed across generations, a GLM was used with a Gaussian error structure, with the target host plant (wheat versus brome), generation number and their interaction as predictors. As the data showed some degree of heteroscedasticity, the dispersion was also modelled within the glmmTMB framework, allowing for separate estimates of variance for each factor combination.

### (ii) Emigration
To test whether both dispersal and acceptance rates differed between wheat and brome, two separate GLMs were built. In both models, a factor coding host species (i.e. 'wheat' or 'brome') was used as predictor, with a binomial distribution for the response and the logit link function. Thereafter, the effect size for each model, Δ, was calculated, which was defined as the mean difference between the dispersal or acceptance rate estimated

for wheat and brome. To test whether the effect size was significantly different from zero, we derived the distribution of $\Delta$ by simulating posterior distributions of model parameters using the 'simulate' function available in the R package 'glmmTMB'. Ten thousand posterior samples were drawn, forming the resampled distribution of the $\Delta$ statistics, given the observed data. Empirical 95% CIs for $\Delta$s were calculated as their 2.5% and 97.5% quantiles, respectively.

### (iii) Experimental evolution

Using survival analysis, we calculated the persistence time (expressed as the number of generations) of WCM populations subjected to three selection regimes: wheat only, brome only and alternating wheat and brome. Populations with at least one individual mite were considered persistent. A proportional hazards model [63] was used to test whether the survival of populations maintained in these regimes differed. Populations that persisted until the end of the experiment (after 273 days = 39 generations) were coded as 'censored'.

### (iv) Seasonal pattern of host plant infestation

GAMMs were used to test whether seasonal patterns of infestation differed between wheat and brome. For all 318 plant samples (181 samples of wheat and 137 of brome, each consisting of 5–22 shoots), the number of successes (infested shoots) and the number of failures (uninfested shoots) were used as response variables in GAMM modelling using a binomial distribution for the response and logit as a link function. Host plants (wheat or brome) and the day of year (fitted as a smooth function representing seasonal patterns of WCM infestation) were used as predictors. Separate fits were allowed for the seasonal patterns for each host (representing a statistical interaction between host plant species and seasonal patterns). Additionally, to account for spatial variation in WCM prevalence, a Gaussian process smooth with a Matérn covariance function [61] was fitted using geographical coordinates, for each host separately. Moreover, as the data were collected over several years (2007–2014), a year identifier was included as a random factor, which was assumed to be an independent and identically distributed random intercept representing between-year variability in the WCM infestation.

## 3. Results

### (a) Population growth rate

The WCM population growth rate differed significantly between hosts (Wald $\chi_1^2 = 342.9$, $p < 0.0001$) and generations (Wald $\chi_2^2 = 83.0$, $p < 0.0001$). The interaction term was not significant (Wald $\chi_2^2 = 3.4$, $p = 0.1842$). On wheat, the mean growth rate was well above replacement in the second, third and fourth generations. By contrast, on brome, the growth rate did not differ from zero in the second and third generations, and it declined below the replacement rate in the fourth generation (figure 1).

### (b) Emigration

The dispersal rate (i.e. the proportion of individuals dispersed by wind) was significantly higher when mites dispersed towards wheat (14.8%, CI: 14.2–15.3) than when they dispersed towards brome (11.2%, CI: 10.7–11.6; Wald $\chi_1^2 = 115.6$, $p < 0.0001$). The mean difference (the effect size, $\Delta$) was 3.6% and the 95% CIs (2.90–4.32) did not include zero (figure 2$a$).

The acceptance rate (i.e. the proportion of individuals that stayed on the plant leaf) also differed significantly between

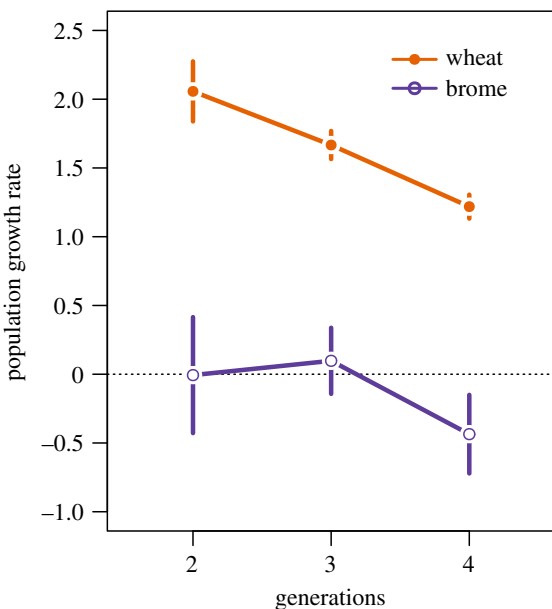

**Figure 1.** WCM population growth rate measured after two, three and four generations on wheat and brome. Points are means and bars are 95% confidence intervals for multiple replicate populations on each plant. (Online version in colour.)

hosts (Wald $\chi_1^2 = 24.48$, $p < 0.0001$) and was higher when mites were placed on wheat (96.5%, CI: 93.3–98.5) than when they were placed on brome (77.6%, CI: 71.4–82.9). The mean difference (the effect size, $\Delta$) was 19.0% and its 95% CIs (13.0–25.5) did not include zero (figure 2$b$).

### (c) Experimental evolution

Smooth brome sustained WCM populations for fewer than 15 generations in all replicates. Mite populations persisted for longer periods on brome when it was temporally interspersed with wheat. However, even in this case, populations could not persist for longer than 22 generations (figure 3$a$). Survival differed significantly among host–plant selection regimes (Wald test $z = 2.40$, $p = 0.0164$). The median persistence time on brome was 4.5 generations (95% CI: 4.5–7.5), whereas, in a fluctuating habitat, it was 7.5 (4.5–13.5) generations. In the wheat regime, no extinctions were recorded (figure 3$b$).

### (d) Seasonal pattern of host plant infestation

The GAMM modelling revealed significant host and spatio-temporal patterns in WCM infestations (table 1). Indeed, there were clear differences in WCM prevalence on both host plants according to the season. On wheat, the prevalence was relatively stable from the beginning of May (which corresponds to wheat emergence) until the end of July, when it started to increase gradually, reaching the highest levels just before harvest. By contrast, brome was virtually uninfested during the time when wheat was available in the fields, but it became infested outside of this period (figure 4).

## 4. Discussion

In this study, we showed that brome is a sink environment for WCMs, because its long-term growth rate in this environment was below the replacement threshold. We also found that mites were less likely to disperse towards and accept brome relative to wheat. Additionally, mite populations failed to

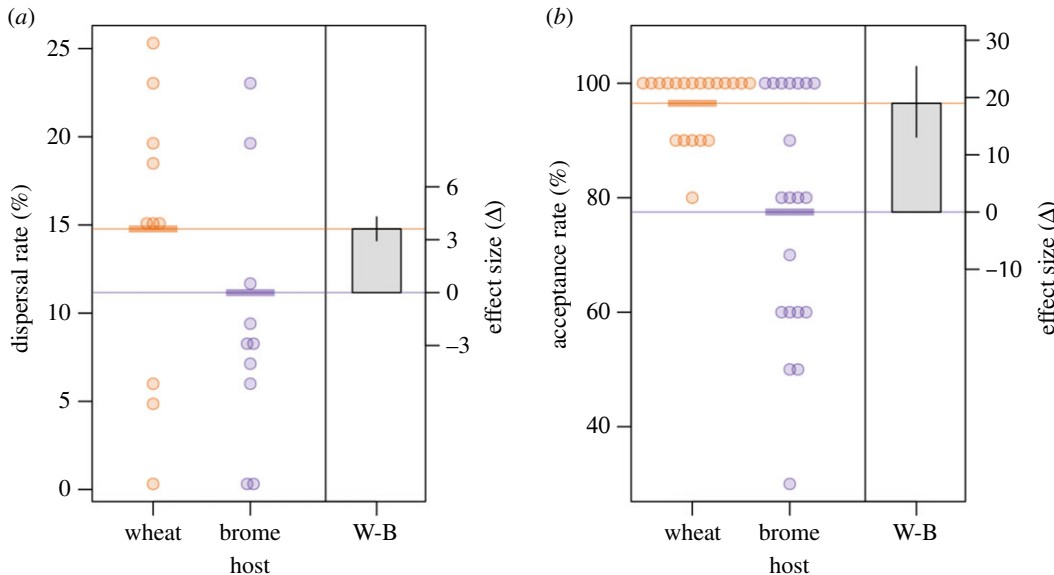

**Figure 2.** Emigration of WCM was measured as (*a*) dispersal rates towards wheat and brome and (*b*) acceptance rates on wheat and brome. Points are the observed values and horizontal lines represent estimated means. The grey box (right panel of each graph) displays the mean difference between both treatments (effect size). The error bars of boxes represent 95% confidence intervals for the effect size. (Online version in colour.)

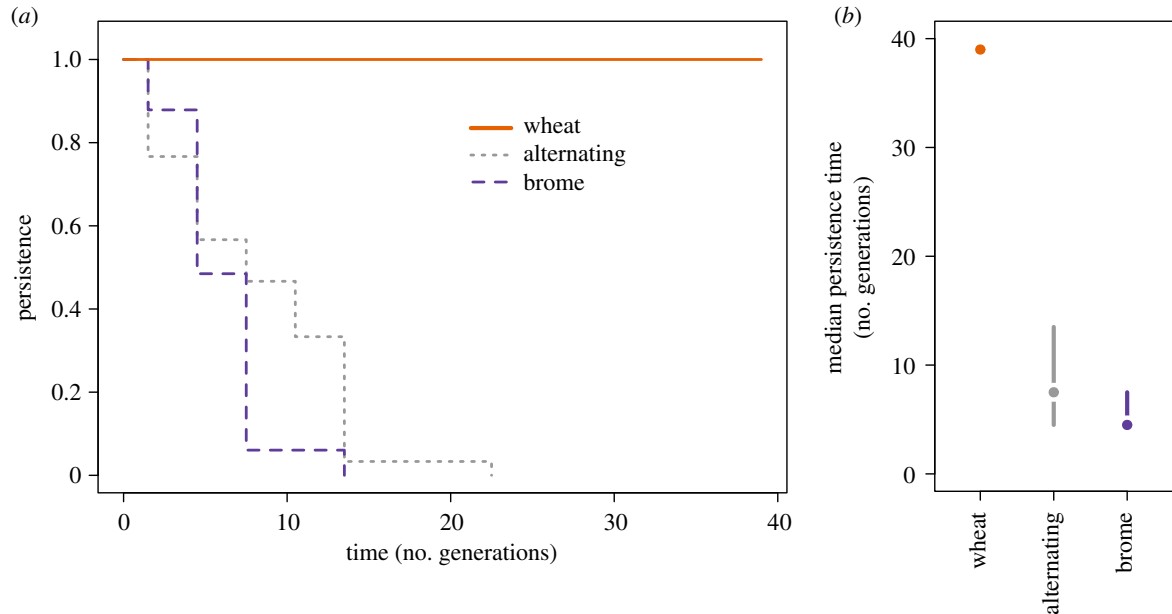

**Figure 3.** Failure of WCM to adapt to brome when experimentally evolving on wheat, brome or alternating on those hosts: (*a*) population persistence (i.e. survival curves estimated for populations, where a population was considered as alive when consisting of at least one individual) and (*b*) median persistence (no. generations) estimated for each regime. (Online version in colour.)

adapt to the brome environment, even in conditions of a heterogeneous environment, i.e. when interspersed with wheat, which should select toward generalists [20]. Finally, we showed that WCM is found on brome in the field only in periods in which wheat is absent. Together, these results show that brome is a sink environment that allows the persistence of WCM populations when the source environment (wheat) is unavailable.

## (a) The failure of evolutionary rescue in the sink environment

Given the ubiquitous occurrence of sink habitats, one possibility is that populations adapt to this environment, thus leading to evolutionary rescue [64,65]. The likelihood of such evolutionary rescue increases with the amount of standing genetic variation available for adaptation to the sink environment [15,66,67]. The WCM population we used to test adaptation to brome was established from a large number of individuals collected at several distant geographical locations, a procedure that is likely to maximize genetic variance [53]. By following this method, we ensured a much larger level of genetic variation than that usually found in populations used in most studies of experimental evolution, which have shown adaptation to a given environment [19,68,69]. Despite many attempts, we did not find that WCM adapted to brome. Therefore, it is unlikely that WCM adaptation to brome occurs in the field, where the populations colonizing brome are probably much smaller than those we used in the laboratory.

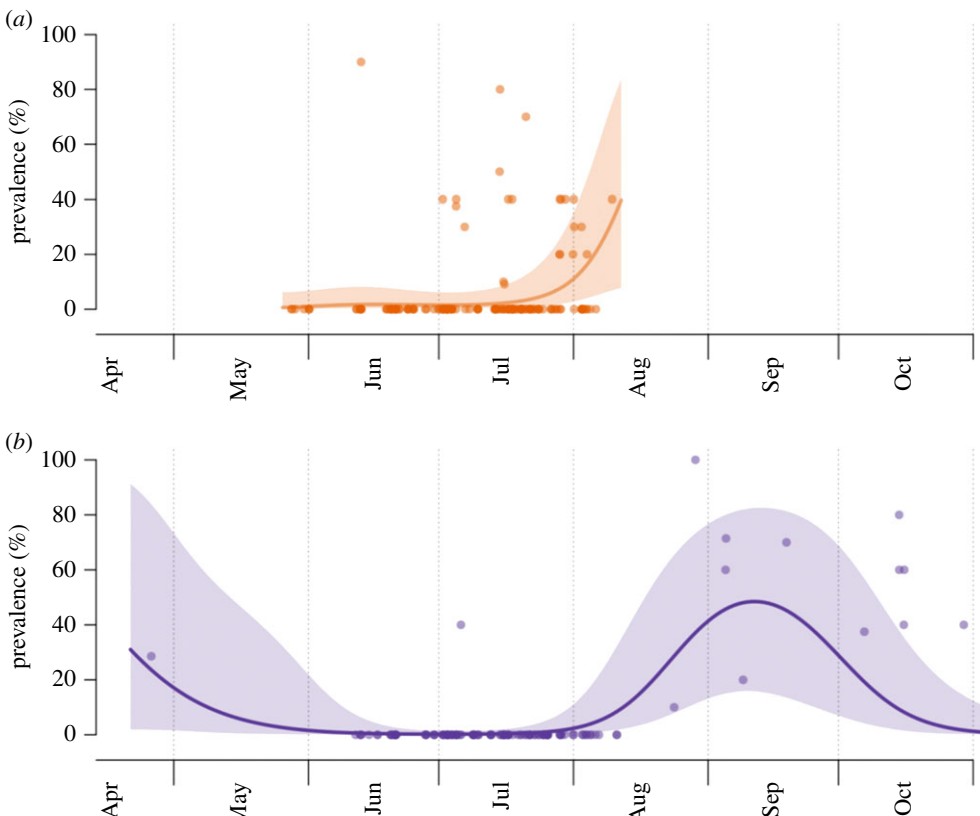

**Figure 4.** Seasonal patterns of WCM prevalence (% of naturally infested shoots) on (*a*) wheat (top panel) and (*b*) brome (bottom panel) based on samples collected from 318 localities in Poland (electronic supplementary material, appendix S4). Solid lines are smooth fits and shaded regions represent 95% confidence bands around these fits. (Online version in colour.)

**Table 1.** Significance of model terms (Wald test) in the generalized additive model examining temporal patterns of natural host plant infestation by WCM. (Estimated degrees of freedom (edf) reflect the smoothness of the fitted curve. The overall model fit, expressed as explained deviance, is 55.3%.)

| model term | edf | Wald statistics | p-value |
|---|---|---|---|
| parametric term: | | | |
| host plant species | 1.0 | 9.8 | 0.0018 |
| smooth terms: | | | |
| seasonal pattern on wheat | 2.8 | 34.1 | <0.0001 |
| seasonal pattern on brome | 4.1 | 70.7 | <0.0001 |
| spatial trend surface on wheat | 5.8 | 34.0 | <0.0001 |
| spatial trend surface on brome | 2.0 | 7.4 | 0.0242 |
| random intercept for year | 5.3 | 61.7 | <0.0001 |

## (b) Does a variable environment lead to the evolution of generalists?

In the laboratory, we tested whether mite populations adapted to brome when placed in an environment where this plant was temporally interspersed with wheat, which roughly corresponded to the conditions they experience in the field. Theory predicts that evolution in such temporally variable environments selects for generalists (e.g. [20,21]). While

some experimental evolution studies have met these predictions [70,71], many others have found different evolutionary outcomes, such as diversification among [28] or within [30] populations or even higher performance than specialists in all environments [25]. Other studies failed to find signs of adaptation to a temporally varying environment [19,26,72]. Here, we also found that experimental evolution with temporal variation between brome and wheat did not result in the ability to use both hosts; rather, this condition led to extinction, albeit at a slower rate than that occurring in a constant brome environment. We also found that mites settled less on brome and tended to move to wheat more frequently than to brome. This form of habitat choice is predicted to hamper adaptation to sink environments [73,74]. Moreover, adaptation to temporal sinks is favoured when the rate of change between environments is slow [16], which was not the case in our laboratory experiment, or the field. On the other hand, temporal auto-correlation is expected to facilitate adaptation to a temporally fluctuating environment [16,75]. This could be the case in our system, as an alternation between good and bad environments occurred every three generations. Whether such alternation selects for a generalist is expected to hinge on the degree of genetic variance for traits associated with adaptation relative to the pace of environmental change [76] and on whether a genetic trade-off between adaptation to the two environments and/or a cost of plasticity exists [21]. We do not have information on these variables in our system. Still, we did not observe the evolution of a generalist via experimental evolution. In the field, WCMs were found on brome only when wheat was absent. Additionally, data on the population growth rate and population prevalence in

the field over several generations showed population declines in brome, despite relatively constant numbers in the initial generations. Moreover, behavioural data showed that WCMs tended to leave brome and colonize wheat more often than the reverse. Hence, these data confirm that brome is clearly a sink environment. Our results emphasize that short-term data on field distribution and population growth rates are not sufficient to ascertain whether a given habitat is part of the fundamental niche of a population because identifying sinks may require data on at least several generations.

## (c) The maintenance of sink habitats

We have clear evidence supporting the fact that the realized niche of WCMs is larger than its fundamental niche, which is one of the most direct consequences of the existence of sinks [7,77]. The evolutionary maintenance of sink habitats is possible when (i) fitness in the source is temporarily lower than fitness in the sink and (ii) individuals that disperse into the sink leave descendants, which can then successfully disperse back into the source [78,79]. Our system fulfils these conditions. Indeed, after harvest, the effective fitness on wheat is zero, whereas on brome, although populations are declining, they can thrive for a few generations, implying that their fitness on this plant is higher than that on wheat. Moreover, as populations in the sink environment persist over a few generations, as evidenced from field and laboratory data, they can recolonize the source once wheat becomes available again. Additionally, our habitat choice experiments show that brome is not an ecological trap, as both the acceptance and the colonization of this habitat by WCMs are lower than those of wheat. The fact that brome is not an attractive sink is expected to favour population persistence [80,81].

## (d) Sinks as temporal refuges

Interest in source–sink systems has generally focused on their spatial dimensions [82], in which sinks may stabilize metapopulations by providing alternative habitats when sources are overcrowded [7]. Temporal aspects of source–sink dynamics have been considered more rarely. For example, Boughton [38] showed, in a metapopulation of the herbivorous butterfly *Euphydras editha*, that similar habitats could be sources or (pseudo)sinks depending on the complex temporal dynamics caused by environmental effects on both plant senescence and the butterfly life cycle. Similarly, Johnson [39] showed that source–sink dynamics in populations of the rolled-leaf beetle *Cephaloleia fenestrate* depend on the frequency of floods.

Our laboratory data suggest that WCM can persist on brome for only a few generations. One may ask whether that is sufficient to overcome the approximately 10-month period (August–May) in which wheat is absent in the field. Generation time in ectotherms increases with a decrease in temperature [83]. Considering the monthly mean temperatures in Poland and the relationship between the temperature and WCM developmental time, we can roughly estimate the expected number of WCM generations produced when fields are without wheat. Based on this, the cumulative number of generations from August to May in natural conditions is estimated to be 5.6 (electronic supplementary material, appendix S5, figures S3–S6),

which roughly fits the number of generations that WCM persisted on brome in our experimental evolution (electronic supplementary material, appendix S5 and figure S7). Our data thus strongly suggest that brome, a sink habitat, can serve as a temporal refuge for WCM populations, potentially allowing them to recolonize wheat, i.e. their source environment, once this plant becomes available again. That is, the sink habitat may act as a source, not via evolutionary rescue, but rather by allowing population persistence in the system despite declining population sizes. In other words, we show that brome serves as a temporal stepping stone for the persistence of this herbivore in the source environment. This potential role of sinks in temporally varying environments has been postulated in mathematical models [79,84,85], but to our knowledge, it has not been specifically tested.

## 5. Conclusion

Together, our results have broad-ranging implications for the understanding of populations' evolutionary responses to varying environments. They emphasize the importance of temporal source–sink dynamics in shaping species' ecological niches and have significant implications for explaining patterns of host use by specialists.

Data accessibility. Data are available from the Zenodo repository: https://zenodo.org/record/4463468.

Authors' contributions. A.L.: conceptualization, data curation, formal analysis, funding acquisition, investigation, methodology, project administration, resources, supervision, validation, writing—original draft, writing—review and editing; S.Ma.: conceptualization, methodology, validation, writing-original draft, writing—review and editing; M.L.: investigation, methodology, resources, writing—review and editing; E.P.: investigation, resources, writing—review and editing; K.K.-I.: investigation, methodology, writing—review and editing; A.R.: investigation, writing—review and editing; S.Me.: conceptualization, writing—original draft, writing—review and editing; L.K.: conceptualization, data curation, formal analysis, methodology, resources, validation, visualization, writing—original draft, writing—review and editing; A.S.: conceptualization, funding acquisition, investigation, methodology, project administration, resources, supervision, validation, writing-original draft, writing—review and editing. All authors gave final approval for publication and agreed to be held accountable for the work performed therein.

Competing interests. We declare we have no competing interests.

Funding. The study was supported by the National Science Centre, Poland (NSC); research grant no. 2017/27/N/NZ8/00305 (tasks: assessing the population growth rate and host acceptance; experimental evolution); NSC research grant no. 2016/21/B/NZ8/00786 (tasks: establishing the stock population—including fieldwork, barcoding and maintaining; assessing the dispersal rate). A.L. scholarships were funded by European Social Funds POWR. 03.02.00-00-I006/17, NSC (PhD scholarship no. 2019/32/T/NZ8/00151) and Adam Mickiewicz University Foundation (awarded in 2019/2020). L.K. was involved in this work while supported by the NSC grant no. 2018/29/B/NZ8/00066. S.Me. became involved in this work while supported by a US Fulbright Scholarship.

Acknowledgements. The authors wish to thank Brian Rector, Katarzyna Kaszewska-Gilas, Łukasz Broda, Agnieszka Majer and Kabita Bharati for their help in field and laboratory work, Jarosław Raubic for DNA barcoding and help with setting dispersal tubes, the company DANKO Hodowla Roślin Sp. z o. o. for the *Triticum aestivum* seeds, the company CENTNAS Sp. z o. o in Krotoszyn, Poland and Botanical Garden in Bydgoszcz, Poland for the *Bromus inermis* seeds. We are also grateful to the Editor and two anonymous referees for their helpful remarks, which greatly improved our manuscript.

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
