## [Peer Review File · Proceedings of the Royal Society B: Biological Sciences]

Review History

RSPB-2021-0665.R0 (Original submission)

Review form: Reviewer 1

Recommendation

Accept with minor revision (please list in comments)

Scientific importance: Is the manuscript an original and important contribution to its field?

Excellent

General interest: Is the paper of sufficient general interest?

Good

Quality of the paper: Is the overall quality of the paper suitable?

Good

Is the length of the paper justified?

Yes

Should the paper be seen by a specialist statistical reviewer?

No

Do you have any concerns about statistical analyses in this paper? If so, please specify them explicitly in your report.

No

It is a condition of publication that authors make their supporting data, code and materials available - either as supplementary material or hosted in an external repository. Please rate, if applicable, the supporting data on the following criteria.

Is it accessible?

Yes

Is it clear?

Yes

Is it adequate?

No

Do you have any ethical concerns with this paper?

No

Comments to the Author

This manuscript presents an interesting study of a mite living on two host plants, which differ in quality. The key result is that the poor-quality host (brome) can act as a temporal refuge when the high-quality host (wheat) is unavailable. They also show that populations were not able to adapt to the sink host and be rescued from extinction. Overall, the manuscript is well written and nicely integrates the results of multiple laboratory experiments and a long-term observational dataset. I have a few critical comments detailed below.

One of my concerns is the mismatch between the timescale at which extinction occurred in the evolution experiments and the timescale at which the natural environment was changing. More specifically, the conclusion that mites can use brome as a temporal refuge in the field despite negative growth rates doesn't match the timescale of extinction in the experiments, which showed these populations on brome only last a few generations. Especially if "the populations colonising brome are probably much smaller than those we used in the laboratory". This deserves more attention if the author's main conclusion about what happens in this system hinges on a population being able to last many months on this poor-quality host in order to recolonize wheat in the spring.

Minor comments:

Line 80 If the author's goal is to evaluate whether the mite is a generalist, it would be helpful to present a definition of a generalist. For example, does a generalist species need to have a positive growth rate on multiple hosts? Does it need to be able to adapt to brome? Does it need to occur on both hosts in the field?

Line 84 Consider rephrasing as "performed an experimental evolution study".

Line 85 The description "with and without immigration from the source" does not seem to match the fluctuating environmental conditions presented below in the methods. From my understanding these are closed populations.

Line 123-127 Was counting destructive? If so, consider stating this as it might clarify the need for 30 pots per species to produce 10 replicates of each species and time.

Line 142 The author's mention that mite density ranged from 1000-3500. Is it possible to test for the effect of this density on the probability of dispersing? One might expect crowding to increase dispersal.

Line 198 Shouldn't this be natural log. Also, if you are defining r as a rate, as in per unit time, I would suggest dividing by the number of days or number of generations that have passed.

Line 300 The results of this study support the idea of brome being a sink habitat. Is there a reason the results here might be so different from the previous study which found an R_0 of 8.2 on brome?

Line 303 It's not been made clear why interspersing with wheat would make it more likely for a population to adapt to brome.

Line 305 This feels like a strong statement. If populations go extinct on brome in a maximum of 12 generations (median of 4.5) and wheat is absent August to May, can it be concluded that brome is responsible for the continued persistence of this species in the field? Perhaps the authors could discuss whether there are additional alternate hosts, or some aspect of the field that slows the population decline.

Line 315 The starting population was created to maximize genetic variation, but the individuals were collected entirely from wheat hosts. Hence if there are particular alleles that allow for adaptation to brome, they may have been missed just due to the initial method of creating the starting population. Was there a reason no individuals were collected from brome?

Line 338 While the switching between wheat and brome in the field may be abrupt, it can act like an autocorrelated environment; wheat is available for many generations then brome is available for many generations, representing a string of good environments and a string of bad environments. Holt et al. 2004a concludes that adaptation is facilitated by autocorrelated environments.

Review form: Reviewer 2

Recommendation

Accept with minor revision (please list in comments)

Scientific importance: Is the manuscript an original and important contribution to its field?

Good

General interest: Is the paper of sufficient general interest?

Good

Quality of the paper: Is the overall quality of the paper suitable?

Excellent

Is the length of the paper justified?

Yes

Should the paper be seen by a specialist statistical reviewer?

No

Do you have any concerns about statistical analyses in this paper? If so, please specify them explicitly in your report.

No

It is a condition of publication that authors make their supporting data, code and materials available - either as supplementary material or hosted in an external repository. Please rate, if applicable, the supporting data on the following criteria.

Is it accessible?

Yes

Is it clear?

Yes

Is it adequate?

Yes

Do you have any ethical concerns with this paper?

No

Comments to the Author

The authors use laboratory experiments and field observations to show that to demonstrate that suboptimal (sink) hosts (brome) allow wheat curl mites (WCM), a herbivorous arthropod pest, to survive in the absence of their preferred host (wheat). They demonstrate that WCM are less likely to disperse toward or remain on the suboptimal host. In the absence of their preferred host, they can persist on the suboptimal host for several generations but not indefinitely, demonstrating that suboptimal/sink habitats can provide an important stepping stone, especially in a seasonal environment, but that evolutionary rescue does not occur.

I very much enjoyed reading this paper. It provides a mixture of evidence to come to strong conclusions. The authors (1) assessed the growth rate on both wheat and brome, (2) measured emigration in terms of both the dispersal rate toward each host and the propensity for WCM females to remain on leaf fragments of each host, (3) found that adaptation to the suboptimal host did not occur and although reproduction was possible, populations eventually died out, and (4) compared seasonal patterns of host plant infestations. Altogether, this is a strong paper showing the temporal importance of sink habitats as stepping stones in seasonal environments. As such, I have surprisingly few comments and they are mostly relatively minor.

- Seasonal pattern of host plant infestation – was each stratum sampled more than once during the season or were they scattered throughout the summer season? If so, how was it determined which strata to sample when?
 - Line 181 states “the total absence of WCM MT-1 genotype”. Were there other genotypes?
 - During the longer period of sampling, were there 37 samples per year or in total? Where the same places sampled each time or different places? How was sampling distributed across the time period? What was the spatial distribution of these samples relative to the larger sampling?
 - The predictors for the emigration GLMs aren’t stated – I assume it’s just wheat versus brome?
 - The geographic coordinates used for the seasonal pattern of host plant infestation GAMMs are not included in the data on zenodo.
 - Figure 3b, is this mean persistence per generation?
 - Figure 4. What exactly is the shading showing?

Copy-edit comments:

- line 27: should be evolutionary rescue, not evolutionarily rescue
- line 80: Should be rigorously evaluating
- line 162: the comma should come after “(three WCM generations at 27C),” not before
- line 182: should it read “... on brome in locations where the main host (wheat) was

available...“?”

- Line 231: there’s an extra “a” in “where used as a response variables”
- line 340: an extra “was” : “WCMs were was found on brome only”

Decision letter (RSPB-2021-0665.R0)

@@date to be populated upon sending@@

Dear Ms Laska:

I am writing to inform you that your manuscript RSPB-2021-0665 entitled "A sink host allows a specialist herbivore to persist in a seasonal source" has, in its current form, been rejected for publication in Proceedings B.

This action has been taken on the advice of referees, who have recommended that substantial revisions are necessary. With this in mind we would be happy to consider a resubmission, provided the comments of the referees are fully addressed. However please note that this is not a provisional acceptance.

- 1) A ‘response to referees’ document including details of how you have responded to the comments, and the adjustments you have made.
- 2) A clean copy of the manuscript and one with 'tracked changes' indicating your 'response to referees' comments document.
- 3) Line numbers in your main document.
- 4) Data - please see our policies on data sharing to ensure that you are complying (<https://royalsociety.org/journals/authors/author-guidelines/#data>).

Sincerely,
 Professor Gary Carvalho
 mailto: proceedingsb@royalsociety.org

Associate Editor
 Comments to Author:

The reviewers both found this manuscript to be a nice combination of laboratory experiments and a long-term observational dataset. The figures and writing are quite nice and the writing is in general very clear.

The major issue is the mismatch between the timescale at which extinction occurred in the evolution experiments and the timescale at which the natural environment was changing. This

time scale issue certainly needs to be acknowledged and discussed. The other major discussion point (perhaps more relevant for readers without entomological backgrounds) degree of control that mites have in their dispersal. How directed is mite dispersal, presumably via wind?

Another detail: reviewer #2 found that the geographical coordinates that are used as part of the GAMM in the analyses are not included in the data, so please insure that the analysis is as reproducible as possible.

Reviewer(s)' Comments to Author:

Referee: 1

Comments to the Author(s)

This manuscript presents an interesting study of a mite living on two host plants, which differ in quality. The key result is that the poor-quality host (brome) can act as a temporal refuge when the high-quality host (wheat) is unavailable. They also show that populations were not able to adapt to the sink host and be rescued from extinction. Overall, the manuscript is well written and nicely integrates the results of multiple laboratory experiments and a long-term observational dataset. I have a few critical comments detailed below.

One of my concerns is the mismatch between the timescale at which extinction occurred in the evolution experiments and the timescale at which the natural environment was changing. More specifically, the conclusion that mites can use brome as a temporal refuge in the field despite negative growth rates doesn't match the timescale of extinction in the experiments, which showed these populations on brome only last a few generations. Especially if "the populations colonising brome are probably much smaller than those we used in the laboratory". This deserves more attention if the author's main conclusion about what happens in this system hinges on a population being able to last many months on this poor-quality host in order to recolonize wheat in the spring.

Minor comments:

Line 80 If the author's goal is to evaluate whether the mite is a generalist, it would be helpful to present a definition of a generalist. For example, does a generalist species need to have a positive growth rate on multiple hosts? Does it need to be able to adapt to brome? Does it need to occur on both hosts in the field?

Line 84 Consider rephrasing as "performed an experimental evolution study".

Line 85 The description "with and without immigration from the source" does not seem to match the fluctuating environmental conditions presented below in the methods. From my understanding these are closed populations.

Line 123-127 Was counting destructive? If so, consider stating this as it might clarify the need for 30 pots per species to produce 10 replicates of each species and time.

Line 142 The author's mention that mite density ranged from 1000-3500. Is it possible to test for the effect of this density on the probability of dispersing? One might expect crowding to increase dispersal.

Line 198 Shouldn't this be natural log. Also, if you are defining r as a rate, as in per unit time, I would suggest dividing by the number of days or number of generations that have passed.

Line 300 The results of this study support the idea of brome being a sink habitat. Is there a reason the results here might be so different from the previous study which found an R_0 of 8.2 on brome?

Line 303 It's not been made clear why interspersing with wheat would make it more likely for a population to adapt to brome.

Line 305 This feels like a strong statement. If populations go extinct on brome in a maximum of 12 generations (median of 4.5) and wheat is absent August to May, can it be concluded that brome is responsible for the continued persistence of this species in the field? Perhaps the authors could discuss whether there are additional alternate hosts, or some aspect of the field that slows the population decline.

Line 315 The starting population was created to maximize genetic variation, but the individuals were collected entirely from wheat hosts. Hence if there are particular alleles that allow for adaptation to brome, they may have been missed just due to the initial method of creating the starting population. Was there a reason no individuals were collected from brome?

Line 338 While the switching between wheat and brome in the field may be abrupt, it can act like an autocorrelated environment; wheat is available for many generations then brome is available for many generations, representing a string of good environments and a string of bad environments. Holt et al. 2004a concludes that adaptation is facilitated by autocorrelated environments.

Referee: 2

Comments to the Author(s)

The authors use laboratory experiments and field observations to show that to demonstrate that suboptimal (sink) hosts (brome) allow wheat curl mites (WCM), a herbivorous arthropod pest, to survive in the absence of their preferred host (wheat). They demonstrate that WCM are less likely to disperse toward or remain on the suboptimal host. In the absence of their preferred host, they can persist on the suboptimal host for several generations but not indefinitely, demonstrating that suboptimal/sink habitats can provide an important stepping stone, especially in a seasonal environment, but that evolutionary rescue does not occur.

I very much enjoyed reading this paper. It provides a mixture of evidence to come to strong conclusions. The authors (1) assessed the growth rate on both wheat and brome, (2) measured emigration in terms of both the dispersal rate toward each host and the propensity for WCM females to remain on leaf fragments of each host, (3) found that adaptation to the suboptimal host did not occur and although reproduction was possible, populations eventually died out, and (4) compared seasonal patterns of host plant infestations. Altogether, this is a strong paper showing the temporal importance of sink habitats as stepping stones in seasonal environments. As such, I have surprisingly few comments and they are mostly relatively minor.

- Seasonal pattern of host plant infestation – was each stratum sampled more than once during the season or were they scattered throughout the summer season? If so, how was it determined which strata to sample when?
- Line 181 states “the total absence of WCM MT-1 genotype”. Were there other genotypes?
- During the longer period of sampling, were there 37 samples per year or in total? Where the same places sampled each time or different places? How was sampling distributed across the time period? What was the spatial distribution of these samples relative to the larger sampling?
- The predictors for the emigration GLMs aren't stated – I assume it's just wheat versus brome?
- The geographic coordinates used for the seasonal pattern of host plant infestation GAMMs are not included in the data on zenodo.
- Figure 3b, is this mean persistence per generation?
- Figure 4. What exactly is the shading showing?

Copy-edit comments:

- line 27: should be evolutionary rescue, not evolutionarily rescue
- line 80: Should be rigorously evaluating
- line 162: the comma should come after “(three WCM generations at 27C),” not before

- line 182: should it read "... on brome in locations where the main host (wheat) was available..."?
- Line 231: there's an extra "a" in "where used as a response variables"
- line 340: an extra "was" : "WCMs were was found on brome only"

Author's Response to Decision Letter for (RSPB-2021-0665.R0)

See Appendix A.

RSPB-2021-1604.R0

Review form: Reviewer 1

Recommendation

Accept with minor revision (please list in comments)

Scientific importance: Is the manuscript an original and important contribution to its field?

Excellent

General interest: Is the paper of sufficient general interest?

Excellent

Quality of the paper: Is the overall quality of the paper suitable?

Excellent

Is the length of the paper justified?

Yes

Should the paper be seen by a specialist statistical reviewer?

No

Do you have any concerns about statistical analyses in this paper? If so, please specify them explicitly in your report.

No

It is a condition of publication that authors make their supporting data, code and materials available - either as supplementary material or hosted in an external repository. Please rate, if applicable, the supporting data on the following criteria.

Is it accessible?

Yes

Is it clear?

Yes

Is it adequate?

Yes

Do you have any ethical concerns with this paper?

No

Comments to the Author

I appreciate the additions of clear predictions if WCM is a generalist (Line 92) and the new analysis of density-dependence (Line 164). The authors have adequately addressed my concern about timescales, reconciling them with a change in temperature (Line 413). Doing so, I believe, strengthens their conclusions. The only comment I still have a concern about, granted a minor one, is that of adaptation to a variable environment (Lines 371- 380). Overall, I don't agree with this paragraph that seems to argue that theory suggests conditions for adaptation aren't optimal. If anything, there is not enough information for theory to make a prediction. More specifically, I don't think the authors can say for sure that this isn't a long enough timescale. Anything more than 2 generations could be enough to favor adaptation (Lyberger et al. 2021 AmNat). And for cyclic environments one must consider the period length relative to the amplitude of oscillations and genetic variance (Lande and Shannon 1996 Evolution).

Decision letter (RSPB-2021-1604.R0)

06-Aug-2021

Dear Ms Laska

I am pleased to inform you that your manuscript RSPB-2021-1604 entitled "A sink host allows a specialist herbivore to persist in a seasonal source" has been accepted for publication in Proceedings B.

The referee(s) have recommended publication, but also suggest some minor revisions to your manuscript. While, as you will see, these are classed as relatively minor, it is important that you address the remaining concern relating to timescale. Therefore, I invite you to respond to the referee(s)' comments and revise your manuscript. Because the schedule for publication is very tight, it is a condition of publication that you submit the revised version of your manuscript within 7 days. If you do not think you will be able to meet this date please let us know.

- 1) A text file of the manuscript (doc, txt, rtf or tex), including the references, tables (including captions) and figure captions. Please remove any tracked changes from the text before submission. PDF files are not an accepted format for the "Main Document".
- 2) A separate electronic file of each figure (tiff, EPS or print-quality PDF preferred). The format should be produced directly from original creation package, or original software format. PowerPoint files are not accepted.

3) Electronic supplementary material: this should be contained in a separate file and where possible, all ESM should be combined into a single file. All supplementary materials accompanying an accepted article will be treated as in their final form. They will be published alongside the paper on the journal website and posted on the online figshare repository. Files on figshare will be made available approximately one week before the accompanying article so that the supplementary material can be attributed a unique DOI.

Sincerely,

Professor Gary Carvalho

Reviewer(s)' Comments to Author:

Referee: 1

Comments to the Author(s).

I appreciate the additions of clear predictions if WCM is a generalist (Line 92) and the new analysis of density-dependence (Line 164). The authors have adequately addressed my concern about timescales, reconciling them with a change in temperature (Line 413). Doing so, I believe, strengthens their conclusions. The only comment I still have a concern about, granted a minor one, is that of adaptation to a variable environment (Lines 371- 380). Overall, I don't agree with this paragraph that seems to argue that theory suggests conditions for adaptation aren't optimal. If anything, there is not enough information for theory to make a prediction. More specifically, I don't think the authors can say for sure that this isn't a long enough timescale. Anything more than 2 generations could be enough to favor adaptation (Lyberger et al. 2021 AmNat). And for cyclic environments one must consider the period length relative to the amplitude of oscillations and genetic variance (Lande and Shannon 1996 Evolution).

Author's Response to Decision Letter for (RSPB-2021-1604.R0)

See Appendix B.

Decision letter (RSPB-2021-1604.R1)

11-Aug-2021

Dear Ms Laska

I am pleased to inform you that your manuscript entitled "A sink host allows a specialist herbivore to persist in a seasonal source" has been accepted for publication in Proceedings B.

Data Accessibility section

Open Access

You are invited to opt for Open Access, making your freely available to all as soon as it is ready for publication under a CCBY licence. Our article processing charge for Open Access is £1700. Corresponding authors from member institutions

Paper charges

Sincerely,

Appendix A

The Editor of the journal Proceedings of the Royal Society B,

Thank you for the opportunity to resubmit our work to Proceedings of the Royal Society B: Biological Sciences. We herewith send you a revised version of the manuscript, based on the comments by the Associate Editor and the Reviewers, which we believe have greatly improved the manuscript. We hope that you will find the manuscript suitable for publication.

Thank you for taking the time to familiarize yourself with our work. I will be pleased to answer any of your questions and await your equitable editorial decision.

Sincerely yours,
Alicja Laska on the behalf of all authors

Associate Editor

Board Member: 1

Comments to Author:

The reviewers both found this manuscript to be a nice combination of laboratory experiments and a long-term observational dataset. The figures and writing are quite nice and the writing is in general very clear.

The major issue is the mismatch between the timescale at which extinction occurred in the evolution experiments and the timescale at which the natural environment was changing. This time scale issue certainly needs to be acknowledged and discussed. The other major discussion point (perhaps more relevant for readers without entomological backgrounds) degree of control that mites have in their dispersal. How directed is mite dispersal, presumably via wind? Another detail: reviewer #2 found that the geographical coordinates that are used as part of the GAMM in the analyses are not included in the data, so please insure that the analysis is as reproducible as possible.

Authors: We are grateful the Associate Editor for his positive comment. Concerning the issues, raised:

We fully agree that the difference in the timescales was not sufficiently acknowledged and we discussed this issue thoroughly in the Discussion (page 22, lines 413-422). We also provided additional data that may explain the discrepancy in time scales via the mite responses to low temperatures, which occur in Poland for large time periods (new Appendix S5 in the Supporting Information).

Indeed, herbivorous mites are passive wind dispersers and have no control on where they land. There is, however, ample evidence that they use several cues when 'deciding' to leave a plant. They are also active ambulatory dispersers, and there is also evidence that such decisions are not random. We added this information explicitly in the Material and Methods section:

"WCMS, as all herbivorous mites, disperse passively with wind, and thus the place where such dispersers land is outside their control (Kennedy and Smitley 1986; Sabelis and Bruin 1996; Michalska et al. 2010). Due to such unpredictability, the

decision to initiate and undertake dispersal is especially crucial. Indeed, there is an ample evidence that herbivorous mites use different types of cues upon which they base their decision to undertake aerial dispersal (Margolies 1993; Jung and Croft 2001). Moreover, they are also ambulatory dispersers, using cues to move from or towards patches (Bitume et al. 2013; Godinho et al. 2020). Here, we evaluate these two types of dispersal in WCM exposed to wheat or brome.” (pages 7-8; lines 143-150). In our opinion this information will be relevant for readers before they become familiar with results.

We added data concerning the geographical coordinates of our study, ensuring that our analysis is reproducible; Zenodo: <https://zenodo.org/record/4463468>.

Reviewer(s)' Comments to Author:

Referee: 1

Comments to the Author(s)

This manuscript presents an interesting study of a mite living on two host plants, which differ in quality. The key result is that the poor-quality host (brome) can act as a temporal refuge when the high-quality host (wheat) is unavailable. They also show that populations were not able to adapt to the sink host and be rescued from extinction. Overall, the manuscript is well written and nicely integrates the results of multiple laboratory experiments and a long-term observational dataset. I have a few critical comments detailed below.

Authors: We are grateful to the Reviewer for this positive opinion. Below we are responding point-by-point to all critical comments. All remarks and suggestions were included in the revised version of the manuscript.

Reviewer 1: One of my concerns is the mismatch between the timescale at which extinction occurred in the evolution experiments and the timescale at which the natural environment was changing. More specifically, the conclusion that mites can use brome as a temporal refuge in the field despite negative growth rates doesn't match the timescale of extinction in the experiments, which showed these populations on brome only last a few generations. Especially if “the populations colonising brome are probably much smaller than those we used in the laboratory”. This deserves more attention if the author's main conclusion about what happens in this system hinges on a population being able to last many months on this poor-quality host in order to recolonize wheat in the spring.

Authors: We are grateful to the Reviewer for this comment. We agree that we had not given sufficient attention to this issue and in the revised version we added a whole paragraph as an explanation that “brome serves as temporal refuge...” as follows: “Our laboratory data suggest that WCM can persist on brome for only a few generations. One may ask whether that is sufficient to overcome the approximately 10-month period (August to May) in which wheat is absent in the field. Generation time in ectotherms increases with a decrease of temperature (Atkinson and Sibly 1997). Considering the monthly mean temperatures in Poland and the relationship

between the temperature and WCM developmental time we can roughly estimate the expected number of WCM generations produced when fields are without wheat. Based on this, the cumulative number of generations from August to May in natural conditions is estimated to be 5.6 (Appendix S5, Figs. S3–S6 in the Supporting Information), which roughly fits the number of generations that WCM persisted on brome in our experimental evolution (Appendix S5, Fig. S7 in the Supporting Information).” (page 22, lines 413-422).

Basically, we believe that the very low temperatures experienced in Poland in the field may help reconcile these different timescales and we provided data and results in the supplements corroborating this. These data and results include: monthly mean temperatures in Poland, WCM generation time in relation to the temperature, expected number of WCM generations in every month, cumulative number of generations until August, and population persistence of WCM on brome (new Appendix S5 in the Supporting Information).

Minor comments:

Reviewer 1: Line 80 If the author’s goal is to evaluate whether the mite is a generalist, it would be helpful to present a definition of a generalist. For example, does a generalist species need to have a positive growth rate on multiple hosts? Does it need to be able to adapt to brome? Does it need to occur on both hosts in the field?

Authors: We agree that the generalist should be more precisely defined. We had mentioned the generalist earlier, when we use this term for the first time (page 4, line 57). In the revised version, we expanded this definition and provided an additional reference (Devictor et al. 2010; page 4, lines 59-60). In the section referred to by the Referee, we specified the criteria upon which we would define that WCM is a generalist based on our data (page 5, lines 92-96). We agree that this adds to the clarity of the paper.

Reviewer 1: Line 84 Consider rephrasing as “performed an experimental evolution study”.

Authors: Changed, as suggested: “Second, we performed an experimental evolution study [...]” (page 5, line 90).

Reviewer 1: Line 85 The description “with and without immigration from the source” does not seem to match the fluctuating environmental conditions presented below in the methods. From my understanding these are closed populations.

Authors: Indeed. We rephrased this statement to be consistent with the methods: “Second, we performed an experimental evolution study in a potential sink or in a source, either with or without temporal variation between these habitats” (page 5, lines 89-91).

Reviewer 1: Line 123-127 Was counting destructive? If so, consider stating this as it might clarify the need for 30 pots per species to produce 10 replicates of each species and time.

Authors: Yes, the counting was destructive, and the clarification was needed. We added: "Since accurately counting the number of mites required destructive sampling, we used 30 pots per host species to obtain 10 replicates per plant species and time interval." (page 7, lines 139-141).

Reviewer 1: Line 142 The author's mention that mite density ranged from 1000-3500. Is it possible to test for the effect of this density on the probability of dispersing? One might expect crowding to increase dispersal.

Authors: As the Reviewer requested we made tests to check whether the dispersal was density dependent. This was based on the assumption that if dispersal rate is modified by population size, there should be a non-linear relationship between the number of individuals that remained (or the no. that dispersed, which is a complementary set) vs. the total no. of individuals in the population. If dispersal is independent of population density, then a constant proportion of individuals will stay, independent of population size, which will result in a straight line describing this relationship (with the slope being just a mean proportion of residents in the whole population). If there is a density dependence, this line will be convex (higher densities incite dispersal, so fewer individuals stay) or concave (higher densities suppress dispersal – more individuals stay).

We tested this idea by relating the number of individuals that remained to the total number of individuals at the start of experiment and fitting a power function to this relationship. Then we tested whether the exponent of the power function was statistically different from one. We expected that the more crowded the population, the fewer animals would stay (so, the proportion of individuals staying should decrease with density resulting in a convex function). We felt that it was less likely that the function would be concave, meaning that the more crowded the population, the more likely it would be for animals to stay (proportion of staying would increase with density). If the value of exponent equals one, the proportion of residents is constant and independent of population density.

The fitted exponent was 0.90 (95% CI: 0.77-1.03). The 95% confidence intervals include 1.0, which means that the null hypothesis cannot be rejected (at the $\alpha=0.05$ level) and that the relationship does not deviate from linearity.

The line of evidence presented above suggests that dispersal is not affected by population density. However, even if we were to assume that dispersal really is density-dependent, this will affect our results only if the population numbers across experimental groups differ which was not the case. On all source (wheat) plants, the numbers of individuals at the start of experiment were similar (Fig. S1) and not statistically different from each other (quasi Poisson GLM: $t=0.29$, $p=0.776$) regardless whether wheat or brome were the target plants.

Fig. S1. Initial population size on source (wheat) plants when wheat or brome were the target plants.

In the Material and Methods section we added text explaining the reasoning and statistical checks described above: “WCM densities on source plants were similar in treatments with both target plant species, and had no effect on dispersal rates (Appendix S3 in the Supporting Information).” (page 8, lines 164-166)

Reviewer 1: Line 198 Shouldn't this be natural log. Also, if you are defining r as a rate, as in per unit time, I would suggest dividing by the number of days or number of generations that have passed.

Authors: We are sorry for this mistake, the formula was indeed \ln , not \log . This has been corrected. Also, the equation was corrected according to the Referee's suggestion – we divided the $\ln(n/n_0)$ by the number of generations. This changes slightly the numerical results, but without any effect on the reasoning and conclusions. Relevant corrections were introduced in the Material and Methods section (formula = $\ln\left(\frac{n}{n_0}\right) / t$, page 11, line 227 and 230), Results section (page 13, lines 273-275), and in Fig. 1.

Reviewer 1: Line 300 The results of this study support the idea of brome being a sink habitat. Is there a reason the results here might be so different from the previous study which found an R_0 of 8.2 on brome?

Authors: The differences between this and previous study arise from differences in study design of both experiments. Specifically:

- (i) different initial sample sizes. In the 2013 study, the experiment was begun with 15 females whereas in this experiment we used 300 individuals (so, $15 \text{ vs } 300 = 20$ times fewer individuals than in previous study). Thus, in the

previous experiment the high initial population growth rate may have been enabled because there were no density-dependent factors, or they appeared later.

- (ii) different temperatures used for incubation and, therefore, number of generations over which R0 was calculated: 20-21°C in the previous study versus 27°C in this study.
- (iii) In this study we assessed the population growth rate of 2nd, 3rd and till 4th generations, and it was in these later generations that we observed reduced population growth. In previous study we assessed the population growth rate of the 1st generation (that is G1 alone is not sufficient to see the population decline).

Thus, considering (i), (ii) and (iii), R0 in the two studies are not directly and straightforwardly comparable.

Reviewer 1: Line 303 It's not been made clear why interspersing with wheat would make it more likely for a population to adapt to brome.

Authors: We clarified this issue as follows:

“Additionally, mite populations failed to adapt to the brome environment, even in conditions of a heterogeneous environment, i.e. when interspersed with wheat, which should select toward generalists (Levins 1968).” (pages 18-19, lines 335-338).

Reviewer 1: Line 305 This feels like a strong statement. If populations go extinct on brome in a maximum of 12 generations (median of 4.5) and wheat is absent August to May, can it be concluded that brome is responsible for the continued persistence of this species in the field? Perhaps the authors could discuss whether there are additional alternate hosts, or some aspect of the field that slows the population decline.

Authors: This comment joins the first comment of Reviewer 1, also referred to by the Associate Editor. We totally concur with this and in the Discussion we put more attention to the timescales at which extinction occurred in the experiment versus the changes in natural environment. We added extra data and text to deal with it (see our reply to the first comment of Reviewer 1 and changes in the manuscript (page 22, lines 413-422 in the Manuscript and Appendix S5 in the Supporting Information).

Reviewer 1: Line 315 The starting population was created to maximize genetic variation, but the individuals were collected entirely from wheat hosts. Hence if there are particular alleles that allow for adaptation to brome, they may have been missed just due to the initial method of creating the starting population. Was there a reason no individuals were collected from brome?

Authors: The stock colony has been established from the material collected during extensive field investigation across the whole country including all possible areas of WCM MT-1 occurrence selected on the base of previous studies: Kuczyński et al. 2016, Skoracka et al. 2017. We visited 85 locations and sampled wheat, barley, triticale and smooth brome. The WCM MT-1 mites were found only on wheat what

supports results of field observations that the mite infests only its preferred host when it is available. Therefore, we assumed that the genetic variation in our stock colony reflects that which is present in the field. The detail description of data collection is available in Appendix S1 in the Supporting Information.

Reviewer 1: Line 338 While the switching between wheat and brome in the field may be abrupt, it can act like an autocorrelated environment; wheat is available for many generations then brome is available for many generations, representing a string of good environments and a string of bad environments. Holt et al. 2004a concludes that adaptation is facilitated by autocorrelated environments.

Authors: Yes, that is correct. Please, note, however, that (1) the same does not apply to our experimental evolution selection regime, as alternation occurred every three generations and (2) in any case, this did not lead to adaptation to the sink environment. We developed this issue and added respective reference (Peniston et al. 2020) in the Discussion:

“In addition, temporal auto-correlation is expected to facilitate adaptation (Holt et al. 2004a; Peniston et al. 2020). This may be the case in our system, but not on a long enough time scale, as alternation between good and bad environments occurred every three generations. Together, these predictions suggest that conditions for the evolution of a generalist may not be optimal in our system, and may explain why we did not observe the evolution of a generalist via experimental evolution.” (page 20, lines 374-380).

Referee: 2

Comments to the Author(s)

The authors use laboratory experiments and field observations to show that to demonstrate that suboptimal (sink) hosts (brome) allow wheat curl mites (WCM), a herbivorous arthropod pest, to survive in the absence of their preferred host (wheat). They demonstrate that WCM are less likely to disperse toward or remain on the suboptimal host. In the absence of their preferred host, they can persist on the suboptimal host for several generations but not indefinitely, demonstrating that suboptimal/sink habitats can provide an important stepping stone, especially in a seasonal environment, but that evolutionary rescue does not occur.

I very much enjoyed reading this paper. It provides a mixture of evidence to come to strong conclusions. The authors (1) assessed the growth rate on both wheat and brome, (2) measured emigration in terms of both the dispersal rate toward each host and the propensity for WCM females to remain on leaf fragments of each host, (3) found that adaptation to the suboptimal host did not occur and although reproduction was possible, populations eventually died out, and (4) compared seasonal patterns of host plant infestations. Altogether, this is a strong paper showing the temporal importance of sink habitats as stepping stones in seasonal environments. As such, I have surprisingly few comments and they are mostly relatively minor.

Authors: We are grateful for all the Reviewer 2 comments and for favourable opinion. We applied all remarks, which were very helpful. We respond to Reviewer's comments below.

Reviewer 2: Seasonal pattern of host plant infestation – was each stratum sampled more than once during the season or were they scattered throughout the summer season? If so, how was it determined which strata to sample when?

Authors: Each stratum was sampled once during the season. We added this information and explanation to Material and Methods section (page 10, lines 199-202)

Reviewer 2: Line 181 states “the total absence of WCM MT-1 genotype”. Were there other genotypes?

Authors: Yes, other WCM genotypes that are specialised to smooth brome were present, namely MT-9, MT-10 and MT-14. This confirmed that the absence of MT-1 was not due to the characteristics of the plants (e.g. their unsuitability for eriophyid mites in general) or any field characteristics such as remoteness or general environmental unsuitability. We added the information that other genotypes were found on brome: “Other WCM genotypes specialised to brome, namely MT-9, MT-10 and MT-14 (Skoracka et al. 2018) were found on brome in these locations.” (page 10, lines 210-211).

Reviewer 2: During the longer period of sampling, were there 37 samples per year or in total? Where the same places sampled each time or different places? How was sampling distributed across the time period? What was the spatial distribution of these samples relative to the larger sampling?

Authors: There were 37 samples in total, collected from different places. Sampling was made from April to October. This information is provided on the pages 10-11, lines 213-214. The spatial distribution of both sampling was clarified by marked the sampling localities in different colours in Fig. S2 in Appendix S4 in the Supporting Information.

Reviewer 2: The predictors for the emigration GLMs aren't stated – I assume it's just wheat versus brome?

Authors: Yes, exactly. We added this clarification: “In both models a factor coding host species (i.e. “wheat” or “brome”) was used as predictor [...]” (page 12, lines 240-241).

Reviewer 2: The geographic coordinates used for the seasonal pattern of host plant infestation GAMMs are not included in the data on zenodo.

Authors: We added the coordinates of the data set and uploaded them to Zenodo repository under: <https://zenodo.org/record/4463468>.

Reviewer 2: Figure 3b, is this mean persistence per generation?

Authors: Thank you for pointing out this issue. After giving it some thought, we restructured the panel b on the Fig. 3. We agree that the term “mean persistence” was misleading and difficult to interpret. Now, the panel b shows the median

persistence time for each treatment, which is more straightforward. The figure legend was corrected (page 16, lines 309-312).

Reviewer 2: Figure 4. What exactly is the shading showing?

Authors: The shaded area corresponds to 95% confidence intervals. We added this information in the figure caption (page 17, lines 326-327).

Copy-edit comments:

Reviewer 2: line 27: should be evolutionary rescue, not evolutionarily rescue

Authors: Corrected (page 2, line 28).

Reviewer 2: line 80: Should be rigorously evaluating

Authors: Corrected (page 5, line 86).

Reviewer 2: line 162: the comma should come after “(three WCM generations at 27C),” not before

Authors: Corrected (page 9, line 186).

Reviewer 2: line 182: should it read “... on brome in locations where the main host (wheat) was available...”?

Authors: Corrected (page 10, line 209).

Reviewer 2: Line 231: there’s an extra “a” in “where used as a response variables”

Authors: Corrected (page 13, line 260).

Reviewer 2: line 340: an extra “was”: “WCMs were was found on brome only”

Authors: Corrected (page 21, line 380).

Appendix B

The Editor of the journal Proceedings of the Royal Society B,

Thank you for the opportunity to publish our manuscript in Proceedings of the Royal Society B: Biological Sciences. We herewith send you a revised version of the manuscript, based on the minor comment made by the Reviewer 1, which we believe have cleared our statement about timescale needed for adaptation in a variable environment.

Sincerely yours,
Alicja Laska on the behalf of all authors

Reviewer(s)' Comments to Author:

Referee: 1

I appreciate the additions of clear predictions if WCM is a generalist (Line 92) and the new analysis of density-dependence (Line 164). The authors have adequately addressed my concern about timescales, reconciling them with a change in temperature (Line 413). Doing so, I believe, strengthens their conclusions. The only comment I still have a concern about, granted a minor one, is that of adaptation to a variable environment (Lines 371- 380). Overall, I don't agree with this paragraph that seems to argue that theory suggests conditions for adaptation aren't optimal. If anything, there is not enough information for theory to make a prediction. More specifically, I don't think the authors can say for sure that this isn't a long enough timescale. Anything more than 2 generations could be enough to favor adaptation (Lyberger et al. 2021 AmNat). And for cyclic environments one must consider the period length relative to the amplitude of oscillations and genetic variance (Lande and Shannon 1996 Evolution).

Authors:

We would like to thank the reviewer for their positive comment. We agree that this part of the text may give the impression to the reader that we are certain that adaptation to a variable environment will not occur in our system. We would like to emphasize that this is not the case. We have now tried to convey a more balanced view of this issue. We thank the reviewer for pointing out these references. We have included the reference from Lande and Shannon (1996). We also took upon ourselves to discuss a few other variables that affect the evolution of generalists based on the seminal paper by Rees Kassen (2002). As for the publication authored by Lyberger et al. (2021), we opted to not include it because it concerns a single event of change, whereas our system represents a temporal sequence of alternative environments.